# Adhesion GPCR Gpr126 (Adgrg6) Expression Profiling in Zebrafish, Mouse, and Human Kidney

**DOI:** 10.3390/cells12151988

**Published:** 2023-08-02

**Authors:** Salvador Cazorla-Vázquez, Peter Kösters, Simone Bertz, Frederick Pfister, Christoph Daniel, Mark Dedden, Sebastian Zundler, Tilman Jobst-Schwan, Kerstin Amann, Felix B. Engel

**Affiliations:** 1Experimental Renal and Cardiovascular Research, Department of Nephropathology, Institute of Pathology, Friedrich-Alexander-Universität Erlangen-Nürnberg (FAU), 91054 Erlangen, Germany; s_cazorla@hotmail.com (S.C.-V.); peter.koesters@fau.de (P.K.); 2Institute of Pathology, University Hospital Erlangen, Friedrich-Alexander-Universität Erlangen-Nürnberg (FAU), 91054 Erlangen, Germany; simone.bertz@uk-erlangen.de; 3Department of Nephropathology, Institute of Pathology, University Hospital Erlangen, Friedrich-Alexander-Universität Erlangen-Nürnberg (FAU), 91054 Erlangen, Germany; frederick.pfister@humanpathologie.com (F.P.); christoph.daniel@uk-erlangen.de (C.D.); kerstin.amann@uk-erlangen.de (K.A.); 4Department of Medicine 1, University Hospital Erlangen, Friedrich-Alexander-Universität Erlangen-Nürnberg (FAU), 91054 Erlangen, Germany; mark.dedden@uk-erlangen.de (M.D.); sebastian.zundler@uk-erlangen.de (S.Z.); 5Department of Nephrology and Hypertension, Friedrich-Alexander-Universität Erlangen-Nürnberg (FAU), 91054 Erlangen, Germany; tilman.jobst-schwan@uk-erlangen.de; 6Research Center On Rare Kidney Diseases (RECORD), University Hospital Erlangen, Friedrich-Alexander-University Erlangen-Nürnberg (FAU), 91054 Erlangen, Germany

**Keywords:** adhesion GPCR, kidney development, Gpr126, parietal epithelial cell, collecting duct, urothelium, ionocytes, zebrafish, RNAscope

## Abstract

Adhesion G protein-coupled receptors (aGPCRs) comprise the second-largest class of GPCRs, the most common target for approved pharmacological therapies. aGPCRs play an important role in development and disease and have recently been associated with the kidney. Several aGPCRs are expressed in the kidney and some aGPCRs are either required for kidney development or their expression level is altered in diseased kidneys. Yet, general aGPCR function and their physiological role in the kidney are poorly understood. Here, we characterize in detail *Gpr126* (*Adgrg6*) expression based on RNAscope^®^ technology in zebrafish, mice, and humans during kidney development in adults. *Gpr126* expression is enriched in the epithelial linage during nephrogenesis and persists in the adult kidney in parietal epithelial cells, collecting ducts, and urothelium. Single-cell RNAseq analysis shows that *gpr126* expression is detected in zebrafish in a distinct ionocyte sub-population. It is co-detected selectively with *slc9a3.2*, *slc4a4a*, and *trpv6*, known to be involved in apical acid secretion, buffering blood or intracellular pH, and to maintain high cytoplasmic Ca^2+^ concentration, respectively. Furthermore, *gpr126*-expressing cells were enriched in the expression of potassium transporter *kcnj1a.1* and *gcm2*, which regulate the expression of a calcium sensor receptor. Notably, the expression patterns of *Trpv6*, *Kcnj1a.1*, and *Gpr126* in mouse kidneys are highly similar. Collectively, our approach permits a detailed insight into the spatio-temporal expression of *Gpr126* and provides a basis to elucidate a possible role of Gpr126 in kidney physiology.

## 1. Introduction

The adhesion G protein-coupled receptor (aGPCR) family comprises 33 members in humans (31 mouse and 24 zebrafish orthologs). The hallmark of aGPCRs is their multitude of adhesion-like domains in a large N-terminus and their unique autocatalytic processing [1]. All, except for one member, contain in their extracellular portion a GAIN domain with a GPCR proteolysis site (GPS) motif. Receptor cleavage generates an N-terminal fragment (NTF) that includes most of the extracellular region and a remaining C-terminal fragment (CTF) containing a 7-transmembrane domain (7TM). Functionally, GPCRs require the interaction of a heterotrimeric G protein on the cytoplasmic side of the cell membrane to conduct their biological response. This response is usually mediated by increased intracellular production of cAMP, which initiates a signaling cascade via the activation of cAMP-dependent protein kinases. Crystallography studies of aGPCRs have described cryptic sequences between GAIN domain β-sheets that could act as tethered agonists for CTF activation. These sequences are called Stachel sequences and might be unmasked by conformational changes in the extracellular domain or by dissociation of the cleaved NTF from the CTF [2,3,4]. In addition, NTFs can be shed off and induce non-cell autonomous signals in neighboring or distant targets [2,5].

aGPCRs have been reported to be important for many pathological processes, including inflammatory conditions [6], tumorigenesis events [7], and lung as well as kidney diseases [8,9]. Notably, aGPCRs represent the second-largest class of the GPCR superfamily whereas GPCRs are currently targeted by ~34% of all approved drugs by the US Food and Drug Administration [10]. Therefore, uncovering the function of understudied GPCRs provides a wealth of untapped therapeutic potential. However, their activation mechanisms, ligands, and downstream signaling remain poorly understood. Thus, research on these receptors is required to unravel their physiological relevance and to expose their clinical potential as pharmaceutical targets.

While a plethora of descriptive data indicate that several aGPCRs are expressed in the kidney from zebrafish to humans [1,8,11], little is known about the role of aGPCRs in kidney development. Recent evidence indicates that aGPCRs can control cell polarity [12], mitotic spindle orientation [13], cell migration [12,14], cell aggregation [15], and the transduction of mechanical stimuli [16,17,18]. These are all processes that also play an important role in kidney ontogenesis and, when disturbed, lead to kidney disease [19,20,21,22,23,24]. The best-characterized aGPCR in kidney development is Celsr1 (Adgrc1) [25]. Mice carrying mutations in *Adgrc1* exhibit UB branching defects at embryonic day (E) 13.5. At E17.5, Adgrc1 mutant mice were characterized by kidney growth retardation, dilated cortical tubules, and mitotic spindle misorientation. Adgrf5-knockout (KO) mice exhibited an accumulation of vacuolar-type ATPase pumps at the apical membrane of A-ICs of the collecting ducts, resulting in an excessive acidification of the urine and a metabolic alkalosis. *Adgrf5/Adgrl4* double KO mice exhibit loss of endothelial fenestration and fusion of podocyte foot processes resulting in proteinuria, uremia, and death within 3 months [26,27]. Adgrg3 KO mice in the AKI model exhibited significantly less renal injury compared to wildtype (WT) mice [28]. Notably, Adgrg3 is upregulated in the kidneys from patients with biopsy-proven acute tubular necrosis compared with healthy controls.

Recently, we have observed that Gpr126 (Adgrg6) is expressed in the kidney based on a *lacZ* reporter mouse line [17]. In addition, a database analysis had revealed that a variety of kidney diseases are associated with altered expression of aGPCR including Gpr126 [8]. Gpr126 possesses a signal peptide, a 7TM domain homologous to secretin-like GPCRs, a GAIN domain, GPS motif, an extended N-terminus containing a Complement, Uegf, Bmp1 (CUB), a pentraxin (PTX) and a hormone-binding domain, and 27 putative N-glycosylation sites. Gpr126 is cleaved at the GPS motif in the endoplasmic reticulum. The resulting NTF and CTF locate to the plasma membrane where they are non-covalently associated with each other [29,30]. In silico analysis indicated that the cytoplasmic domain of CTF contains potential sites for palmitoylation; phosphorylation sites for cAMP dependent kinases/protein kinase G, protein kinase C, and casein kinase II; a potential myristoylation signal; and a microbodies C-terminal targeting motif [31]. KO-based phenotypes support a model in which Gpr126 elevates intracellular cAMP through interactions with Gα_s_ [32,33,34]. Chimeric G-protein assays further indicated that GPR126 can also couple to G_i_-proteins [33]. In addition, laminin-211 [35], type IV collagen [36], type VI collagen [37], and the cellular prion protein PrP(C) [38] have been identified as potential Gpr126 ligands that activate G protein signaling. Furthermore, Gpr126 contains in its CTF a short peptide sequence (termed “Stachel” sequence) that functions as a tethered agonist [4,39]. Yet, the consequence of Gpr126-mediated G protein signaling remains poorly understood. Notably, increase in cAMP has also been detected when shaking forces were applied to *Gpr126*-transfected cells [35]. The interaction of Gpr126 with laminin-211 leads to a synergistic increase in intracellular cAMP levels under dynamic conditions (vibration or shaking). Moreover, *Gpr126* expression was found in a variety of tissues known to be subjected to flow or mechanical forces, such as the endocardium of the heart and the hyaline cartilage of the trachea [17].

Gpr126 has been reported to be required for the development of several tissues/organs including the peripheral nervous system, intervertebral disc, ear, placenta, and heart [5,32,34,40,41,42,43]. Yet, despite its clinical importance, little is known about Gpr126-mediated cellular processes and signaling, and thus large efforts are invested in understanding the functional role of Gpr126. Here, we aimed to determine its cellular expression pattern during kidney development in order to provide a basis to elucidate a possible role of GPR126 in kidney function.

## 2. Materials and Methods

### 2.1. Human Material

The data obtained in this study were processed according to the principles of scientific practice and all patient samples were analyzed anonymously. Human embryonal kidney tissues were obtained from routine autopsies after confirming the absence of a morphologically recognizable renal phenotype. Normal kidney and bladder tissues without evidence of disease were obtained from distant portions of kidneys or bladders surgically excised because of the presence of a localized neoplasm and diagnosed at the Institute of Pathology, FAU. Kidney and bladder sections showed normal histology without signs of disease. All tissues were provided as formalin-fixed, paraffin-embedded (FFPE) sections of 4 µm thickness. The use of FFPE tissue material was approved by the Ethics Committee of the Friedrich-Alexander-University of Erlangen-Nürnberg for the archive of the Department of Nephropathology (Re.-No.4415 and Re.-No.22-150-D) and Institute of Pathology (Re.-No. 4607) waiving the need of informed consent due to its retrospective nature.

### 2.2. Animal Housing and Experimentations

Kidneys from wildtype C57BL/6 and *lacZ*-Gpr126 reporter (*Adgrg6tm1a^(EUCOMM)Hmgu^*) mice [17] were used to determine the endogenous expression of Gpr126. Wildtype AB and *Tg(chd17:GFP)* zebrafish strains [44] were utilized to identify epithelium of the pronephric and mesonephric tubules and ducts. The transgenic *cdh17*-GFP line was kindly provided by Prof. Neil Hukriede. Housing, breeding, as well as organ perfusion and extraction conformed with the guidelines from directive 2010/63/EU of the European Parliament of the protection of animals used for scientific purposes and were approved by the local governmental animal protection committee. For experimental details, see below (Re. No. TS-05/11 Pathologie). For LacZ staining, kidneys from *Adgrg6tm1a^(EUCOMM)Hmgu^* mice were obtained as previously described [17]. In brief, mice were anesthetized by inhalation of 3% isoflurane and subjected to transcardial perfusion with 10% dextran 40 supplemented with 0.1% procaine followed by 0.9% NaCl to clear residual blood from tissues (protocol TS-05/11 Pathologie). For RNAscope^®^ in situ hybridization, wildtype C57BL/6 mice were, after isoflurane sedation, transcardially perfused with 0.9% NaCl (protocol TS-05/11 Pathologie). Organs were extracted and post-fixed by immersion in 4% buffered formalin for 24 h at RT before embedding in paraffin.

### 2.3. Antibodies and RNA Probes

The primary antibodies used in this study were anti-cTnI (1:300; Abcam, ab56357), anti-Krt8 (1:6; Developmental Studies Hybridoma Bank, TROMA-I), anti-Claudin-1 (1:100; Abcam, Cambridge, UK, ab15098), anti-E-cadherin (1:200; BD Transduction, BD 610181), anti-Aquaporin-2 (1:100; Sigma-Aldrich, St. Louis, MO, USA, SAB5200110), and anti-V-ATPase B1 (1:100; Invitrogen, PA5-56878). The secondary antibodies were Alexa Fluor^®^ 488 donkey anti-rabbit (1:200; Life Technologies, Carlsbad, CA, USA, A21206) and Alexa Fluor^®^ 647 donkey anti-mouse (1:200; Life Technologies, A31571). The RNA probes targeting Gpr126 in zebrafish, mouse, and human tissues were designed and provided by Advanced Cell Diagnostics^®^ (Dr-adgrg6, 564461; Mm-Adgrg6, 472251; Hs-ADGRG6, 480121).

### 2.4. LacZ Staining

Organs were washed in cold phosphate-buffered saline (PBS), embedded in Tissue-Tek^®^ O.C.T. (*Sakura*, 4583), and frozen in the vapor phase of liquid nitrogen. Samples were stored at −80 °C for a minimum of 24 h and sectioned in 10 µm slices. Fixation was performed with 0.2% glutaraldehyde solution (Sigma-Aldrich, G5882) for 40 min on ice and incubated in a X-gal staining solution (AppliChem, Darmstadt, Germany, 74134) at 37 °C overnight. Sections were counterstained with nuclear fast red (Carl Roth, Karlsruhe, Germany, N069.1) for 5 min at room temperature (RT) and washed in dimethyl sulfoxide (Sigma-Aldrich, D4540) and xylol-free clearing solution (Merck Millipore, Burlington, MA, USA, 1098435000).

### 2.5. Immunofluorescence Staining

FFPE sections with a thickness of 4 µm were deparaffinized in xylol twice for 10 min each at RT. Then, rehydration of the tissue was performed by washing for 5 min in ethanol solutions with gradually decreased concentrations (100%, 96%, 70%, and 50%) and distilled water. To unmask epitopes possibly altered by the fixation process, sections were incubated in a retrieval solution provided by Advanced Cell Diagnostics^®^ at 95 °C for 15 min in a programmable pressure cooker. Sections were then subjected to the same treatment as that for RNAscope^®^ in situ hybridization (see below). Lastly, sections were permeabilized with 0.3% Triton^®^ X-100 in PBS for 10 min at RT and incubated for 1 h at RT with a blocking buffer containing 3% [*w*/*v*] bovine serum albumin (Carl Roth, 8076.4), 5% [*w*/*v*] goat serum (Biowest, Nuaillé, France, S1810), 20 mM MgCl_2_, and 0.3% [*v*/*v*] Tween-20 in PBS. Primary antibodies were incubated at 4 °C overnight, and fluorophore-conjugated secondary antibodies were incubated at RT for 2 h. Sections were finally counterstained with 0.5 μg/mL DAPI (4′,6′-diamidino-2-phenylindole) in 0.1% NP40/PBS at RT for 10 min to visualize nuclei.

### 2.6. RNAscope^®^ In Situ Hybridization

Both chromogenic and fluorescent RNAscope^®^ in situ hybridizations were performed according to the manufacturer’s instructions. Target probes were designed and provided by Advanced Cell Diagnostics^®^ with the rest of the kit components (RNAscope^®^ 2.5 HD Detection Reagents RED, 322360; RNAscope^®^ Multiplex Fluorescent Detection Reagents v2, 323110). FFPE tissue sections were deparaffinized as explained for immunofluorescence staining and subjected to a provided hydrogen peroxidase solution at RT for 10 min to block endogenous peroxidase activity. To restore antigenicity after fixation, an antigen retrieval solution provided by the manufacturer was used at 95 °C for 15 min in a programmable pressure cooker. The tissue was then rinsed in deionized water and immediately treated with a provided protease solution at 40 °C for 30 min. Apart from the zebrafish probe, which had to be diluted first (1:50 in TSA^®^ buffer), 3–4 drops of the according species-specific target probe were added undiluted to the sections before hybridization was performed at 40 °C for 2 h. From this point, the tissue was incubated in different successive amplifier and signal development solutions depending on the chosen kit. Fluorescent signal development additionally required the incubation with Opal^®^ reagents (1:1500, Akoya Biosciences, Marlborough, MA, USA, FP1488001KT) in TSA^®^ buffer at 40 °C for 30 min. To combine in situ hybridization with immunostaining, primary antibodies were incubated overnight at 4 °C after 1 h blocking with a solution containing 0.5% [*w*/*v*] bovine serum albumin (Carl Roth, Karlsruhe, Germany, 8076.4). Secondary antibodies were then incubated for 30 min at RT. To counterstain the chromogenic assay, hematoxylin (Carl Roth, Karlsruhe, Germany, T865.2) was used at RT for 30 s and followed by running warm tap water for 5 min. To counterstain the tissue in the fluorescent assay, freshly prepared DAPI solution (0.5 μg/mL in 0.1% NP40/PBS) at RT for 10 min, DAPI solution provided in the kit for 30 s at RT, or the combination of DAPI solution (40 μg/mL in H_2_O) with Mowiol mounting medium (40 g Mowiol, Calbiochem, San Diego, CA, USA, 475904 in 200 mL 0.2 M Tris-HCl, pH 8.5; 100 mL glycerin; 8.75 g DABCO, Sigma, D-2522 + 2 mL DAPI solution) was utilized.

### 2.7. Microscopy and Image Analysis

Bright-field images were taken using a Zeiss^®^ Axio Zoom V16 stereoscope or an AMG^®^ Evos, Keyence^®^ BZ-900, or Olympus^®^ BX60 microscope. Confocal images were taken with a Zeiss^®^ 800 LM laser scanning microscope. For routine rapid examination of the slides, a Zeiss^®^ AxioTech Vario 100HD and Olympus^®^ BH2 microscope or a Leica^®^ S6 stereoscope were utilized. Image processing and analysis was performed using Zeiss^®^ ZEN (Blue edition), ImageJ with the Fiji extension package, and Adobe^®^ Photoshop CC 2018. Imported z-stacks were projected into a single plane using maximum intensity projection. Confocal images were acquired as 8-bit at a standard resolution of 1024 × 1024 pixels. Laser intensity and signal amplification parameters were maintained between different conditions of the same experiment. To quantify the RNAscope^®^ signal, a semi-quantitative scoring approach was applied, as recommended by Advanced Cell Diagnostics^®^ guidelines. A score from 0 to 4 was given according to the general number of dots and/or dot clusters found in the tissue after RNAscope^®^ in situ hybridization (0: <1 dot/10 cells; 1: 1–3 dots/cell; 2: 4–9 dots/cell, none or infrequent dot clusters; 3: 10–15 dots/cell and/or <10% dots in clusters; 4: >15 dots/cell and/or >10% dots in clusters).

### 2.8. Single-Cell RNAseq Analysis

Single-cell RNA-sequencing data of zebrafish was retrieved from Zebrahub [45]. All cell type annotations referenced in this study are derived from the original study. Each annotated cell type of interest was normalized individually with the “normalize_total” function and scaled with the “log1p” function using Scanpy v1.9.1 [46] with default parameters. Neighbors were calculated based on all genes using the UMAP (Uniform Manifold Approximation and Projection) algorithm [47] with default settings and, subsequently, a two-dimensional plot was constructed. The connectivity matrix output by UMAP was passed to the Leiden [48] algorithm to calculate communities with a resolution of 1. Differential expression analysis was conducted with MAST [49]. Analysis of differentially expressed genes was performed with FishEnrichr [50,51].

### 2.9. Statistical Analysis

Statistical analysis was carried out using GraphPad^®^ Prism 5. Differences between groups were considered statistically significant when the *p*-value was ≤0.05. The statistical significance of differences between groups was tested using Mann–Whitney U test for data based on the RNAscope^®^ score system comparing the means of two groups of data defined by categorical variables. In case of more than two independent groups of data, one-way analysis of variance (ANOVA) was applied followed by Tukey’s honest significance test (Tukey’s HSD).

## 3. Results

### 3.1. RNAscope^®^ Technology Is Suitable to Detect Gpr126 Expression

After testing all commercially available anti-Gpr126 antibodies utilizing Gpr126 KO mice and Gpr126 mutant zebrafish lines, we could not identify a reliable working antibody in mice or zebrafish. In order to study Gpr126 expression at cellular resolution, RNAscope^®^ technology was used to perform RNA in situ hybridization on FFPE tissues. Previously, *Gpr126* has been demonstrated to be expressed in the endocardium of mouse E11.5 hearts by in situ hybridization [5]. Therefore, E11.5 hearts were used as controls, which are composed of an inner single layer of endocardial cells and several layers of cardiomyocytes. RNAscope^®^ fluorescent signal was observed as punctuated dots lining the lumen of the heart chambers on transversal sections (Figure 1a), suggesting an endocardial location. Subsequently, the RNAscope^®^ in situ hybridization assay was combined with anti-cardiac troponin I (cTnI) immunofluorescence staining, which served as a specific marker of cardiomyocytes (Figure 1b). This analysis confirmed *Gpr126* RNA expression in the cTnI-negative endocardial cells, as previously reported [17]. Hence, it was concluded that RNAscope^®^ technology is a valid method to detect the murine *Gpr126* mRNA.

To evaluate the suitability of the RNAscope^®^ technology in the kidney, adult human and mouse kidney FFPE sections were subjected to RNAscope^®^ analysis (Figure 1c,d). In human FFPE renal biopsies, RNAscope^®^ probes targeting the bacterial 4-hydroxy-tetrahydrodipicolinate reductase (*dapB*) gene were chosen as a negative control. As a positive control, RNAscope^®^ probes targeting the human peptidyl-prolyl cis-trans isomerase B (*PPIB*) gene were utilized. While the bacterial RNA probes are not expected to hybridize in eukaryotic systems, *PPIB* is known to be conserved and expressed at low levels, serving as a control for low expressed genes [52]. The RNAscope^®^ fluorescent assay in human tissue showed no punctuated dotty signal when *dapB*-targeting probes were used (Figure 1c). However, samples exhibited areas of diffuse autofluorescence consistent with the light emission properties of the hemoglobin within erythrocytes, as previously documented [53]. When RNAscope^®^ probes targeting *PPIB* were used, a signal was observed as distinctive dotty pattern in addition to the autofluorescence of red blood cells.

To establish the RNAscope^®^ technology in the mouse kidney, adult kidney samples from a *Gpr126-lacZ* reporter mouse (*Adgrg6^tm1a(EUCOMM)Hmgu^*) [17], which allows determining the *Gpr126* RNA expression pattern by X-gal staining, was utilized as a control (Figure 1d). X-gal staining revealed β-galactosidase activity in tubular structures at the renal medullary papilla. This expression pattern was also obtained when utilizing RNAscope^®^ probes for mouse *Gpr126* RNA.

Taken together, these data indicate that the RNAscope^®^ technology is a reliable method to detect human and mouse *Gpr126* RNA on FFPE sections and is compatible with immunohistochemistry. Notably, erythrocytes are identified as diffuse, larger autofluorescent areas that can be distinguished from the RNAscope^®^ signal, which appears as punctuated dots.

### 3.2. Gpr126 Expression Changes in Early Development from Ubiquitous to Nephron-Enriched

To evaluate the *Gpr126* expression pattern during renal development, embryonic tissues from human and mouse metanephric kidneys were analyzed. Human embryos at 11, 20, and 28 weeks of gestation (wg) and mouse embryonic kidneys from E13.0 and E15.5 wildtype animals were utilized. Notably, metanephroi start to form at around 10 wg in humans or E10.5 in mice with the induction of the ureteric bud [54]. Later in development, the nephrogenesis rate increases exponentially from 18 to 23 wg in humans and from E13.0 to birth in mice [55]. RNAscope^®^ analysis revealed that *Gpr126* expression levels were dependent on the developmental stage in both human and mouse samples (Figure 2).

In humans (Figure 2a,c), dot density analysis revealed that *GPR126* expression at 11 wg was moderate in both stromal (RNAscope^®^ score: 2.4 ± 0.49) and epithelial tubular-forming cells (score 3.0 ± 0.63) (n = 4). At 20 wg, *GPR126* expression was increased in both stromal (score 2.6 ± 0.66) and epithelial cells (score 3.8 ± 0.40) (n = 5). Finally, at 28 wg, *GPR126* expression was markedly decreased in stromal cells (score 0.4 ± 0.49) and epithelial cells (score 2.6 ± 0.49) (n = 3). Notably, *GPR126* expression was not significantly different in epithelial cells comparing 28 wg to 11 wg, whereas stromal *GPR126* expression was significantly decreased (*p* < 0.01). To assess if this spatio-temporal expression pattern is conserved in mice, *Gpr126* RNA in situ hybridization was combined with cytokeratin-8 (Krt8) immunostaining (Figure 2b), which serves as a marker of the ureteric bud-derived epithelium [56]. Similar to the human system, *Gpr126* expression in mice increased in epithelial cells during kidney development from E13.0 to E15.5 (E13.0: score 1.4 ± 0.49; E15.5: score 3.2 ± 0.40, *p* < 0.05) but remained low in stromal cells (E13.0: score 1.0 ± 0.63; E15.5: score 0.8 ± 0.40, not significant) (n = 3).

Collectively, these data indicate that both stromal and epithelial cells express *Gpr126* during human and mouse metanephros development. Yet, *Gpr126* expression is enriched over time in the epithelial cells during active nephrogenesis stages.

### 3.3. Gpr126 Is Enriched in Parietal Epithelial Cells and the Urinary Collecting System in Mice and Humans

Single-cell RNAseq data by Ransick et al. from adult murine renal tissue indicate that *Gpr126* might be expressed in different nephron segments such as the parietal epithelial cells and the collecting duct [57]. To validate the parenchymal expression pattern of *Gpr126*, we performed RNAscope^®^ in situ hybridization on FFPE tissue of wildtype C57BL/6 mice. To be able to identify the cell types showing *Gpr126* signal, the *in situ* hybridization was combined with immunofluorescent antibody staining for different cell-specific markers. In contrast to most of the tubular structures in the renal cortex, the *Gpr126* signal could be detected in the renal corpuscle colocalizing with the tight junction protein claudin 1 that is specifically expressed in the parietal epithelial cells of the Bowman’s capsule [58] (Figure 3, left panels). 

Apart from the parietal epithelial cells, *Gpr126* mRNA could be detected in the collecting ducts, which were labelled by co-staining of E-cadherin and the collecting duct-specific water channel protein aquaporin 2 [59]. Distal tubules, also positive for E-cadherin [60,61], and proximal tubules, characterized by a lower epithelial density of nuclei compared to distal tubules and collecting ducts and almost absent E-cadherin staining [60,61], both showed only very low levels of *Gpr126* expression. Interestingly, the density of signal dots varied throughout the different segments of the collecting duct. *Gpr126* expression first decreased from the cortical collecting duct (CCD) to the outer medullary collecting duct (OMCD) before increasing again in the inner medullary collecting duct (IMCD) and the papillary collecting duct (PCD). Taken together, our findings are consistent with the RNA sequencing data from Ransick et al., with renal parenchymal expression being most abundant in parietal epithelial cells and the collecting duct.

In order to characterize the expression of human *GPR126* in adulthood, renal-specific antibodies were explored in combination with RNA in situ hybridization. *GPR126* expression was detected in healthy adult human kidneys in both glomerular and tubular regions of the nephron, as assessed by RNAscope^®^ analysis (Figure 4). In the glomeruli, *GPR126* RNA expression colocalized with the parietal epithelial cell marker claudin 1. In the renal tubule, *GPR126* expression was observed at low-to-no signal along the proximal and distal tubule segments (Figure 4). However, it was consistently evident in both V-ATPase-positive and V-ATPase-negative tubular cells of the collecting duct, suggesting that *GPR126* is expressed in both intercalated and principal cells. 

The study of *Gpr126* expression was extended beyond the renal parenchyma. For this purpose, human FFPE sections from tumor-free areas of different urinary tract tissues and mouse sections from the renal pelvis were subjected to chromogenic and/or fluorescent RNAscope^®^ analysis (Figure 5). Human *GPR126* expression was detected in the inner surface of the renal pelvis, ureter, and bladder (Figure 5a), suggesting a urothelial expression. Fluorescent RNAscope^®^ analysis on human and mouse renal pelvises revealed that *Gpr126* was predominantly detected at the urothelium in both species, while low-to-no signal was found in the underlying lamina propria (Figure 5b).

Collectively, these data suggest that *Gpr126* is expressed throughout the urothelium of the urinary collecting system in humans and mice.

### 3.4. gpr126 Is Expressed in Ionocytes as well as Pro- and Mesonephros in the Zebrafish

Despite the lack of metanephroi, pronephric and mesonephric nephrons in zebrafish display segment-specific morphological and physiological features and spatio-temporal gene expression that are homologous to the human kidney [62]. *gpr126* expression analysis with RNAscope^®^ combined with anti-GFP immunostaining in transversal FFPE sections from *Tg(cdh17:GFP)* larvae indicates that *gpr126* is expressed in the pronephros as well as ionocytes (Figure 6a), which exhibit analogous functions to collecting duct epithelial cells in the embryonic skin and adult gills of zebrafish [63,64]. The analysis of recently published single-cell RNAseq data describing zebrafish development [45] confirmed that *gpr126* is highly expressed in ionocyte progenitor cells at 14 h and in ionocytes at 5 and 10 days post fertilization (hpf or dpf) (Figure 6b,c and Appendix A). As expected, high *gpr126* expression was, for example, also detected at 10 dpf in the otic vesicle [32], cartilage [40], peripheral nervous system [34], and renal glomerulus (based on our mouse data above) (Appendix A) [1].

Collecting duct cells and ionocytes plays an important role in ion homeostasis. Notably, the ion-transport-related genes *nkcc1* and *kcnq1* were reported to be downregulated in the otic vesicles, a main location of ionocytes, in *gpr126* zebrafish mutants [32]. Previously, four types of ionocytes have been described in zebrafish (**H^+^-ATPase-rich**: *apical H^+^-ATPase (HA, atp6v1aa)*, *NHE3b* (*slc9a3.2*), *Rhcg1* (*rhcgb*), *rhbg*, *AE1b* (*slc4a1b*), *atp1a1a.5*, *ca15a*, *ca2* (*cytosolic CA2-like a)*, *ceacam1*, *gcm2*; **Na^+^-K^+^-ATPase-rich**: *ECaC* (*trpv6*), *PMCA2* (*atp2b2*), *NCX1b* (*slc8a1b*), Na^+^-K^+^-ATPase *(atp1a1a.1)*; **Na^+^-Cl^−^ co-transporter-expressing**: *slc12a10.2*, *atp1a1a.2*, *NBCe1b* (*slc4a4b*); **K^+^-secreting**: *atp1a1a.4*, Kir1.1 (*kcnj1a.1*)) [65,66,67,68]. *ndrg1a* has been described as a general marker of ionocytes and the pronephric duct [69]. Therefore, we analyzed the single-cell RNAseq data in regard to these genes to determine in which ionocyte sub-types *gpr126* expression can be detected (Figure 6d).

At 10 dpf, four distinct clusters of ionocytes could be derived from the single-cell RNAseq data (1, 2, 3, 4) (Figure 6b,c). Cluster 1 contains ionocytes in which exclusively *slc4a1b*, *ca15a*, *rhcgb*, and *rhbg* expression could be detected. In addition, they are characterized by enriched expression of H^+^-ATPase (*atp6v1aa*) and *ca2*. Thus, cluster 1 ionocytes represent H^+^-ATPase-rich ionocytes. Notably, several genes known to be expressed by H^+^-ATPase-rich ionocytes are co-expressed in clusters 1 and 2 (*ceacam1*, *atp1a1a.5*, and *gcm2*), while *slc9a3.2* expression could exclusively be detected in cluster 2 (Appendix A). Expression of genes previously identified as marker genes for Na^+^-K^+^-ATPase-rich ionocytes are selectively detectable in ionocytes at 5 dpf, represented by clusters 5 and 6 (Appendix A). Yet in cluster 5, there are also 10 dpf cells included (Figure 6b). Genes previously identified as marker genes for Na^+^-Cl^−^ co-transporter-expressing ionocytes are exclusively detected in cluster 3 or in both cluster 3 and 4 (Appendix A). In contrast to these 3 ionocyte-types, K^+^-secreting ionocytes could not, based on *atp1a1a.4* and *kcnj1a.1* expression, be identified as a distinct cell population/cluster (Appendix A).

In order to predict a possible function of *gpr126*, we determined in which clusters *gpr126* expression can be detected. Notably, *gpr126* expression was selectively detected at 10 dpf in cluster 2 together with *slc9a3.2*, *slc4a4a*, and *trpv6* expression. In addition, cells of cluster 2 were enriched in *kcnj1a.1* expression and co-expressed a subset of the H^+^-ATPase-rich ionocyte marker genes (*ceacam1*, *atp1a1a.5*, and *gcm2*). Taken together, these data indicate that Gpr126 is expressed in a subset of H^+^-ATPase-rich and possibly Na^+^-K^+^-ATPase-rich ionocytes.

To determine whether other aGPCRs exhibit a similar expression pattern as *gpr126* in ionocytes, we analyzed all genes starting with “adgr” (Appendix A). This analysis revealed that only a few aGPCRs are enriched in a specific cluster (Appendix A and Appendix A). *adgrl3* was the only aGPCR found that exhibits a similar pattern as *gpr126* (Appendix A). In addition, we performed a differential expression analysis, where we compared cells in which *gpr126* was detected versus cells that lack *gpr126* UMIs (Appendix A). Among the differentially expressed genes, we detected three genes that have previously been associated to Gpr126 (Appendix A). It has been described that GPR126 regulates angiogenesis through modulation of VEGFR2 receptor signaling [70]. Further, GPR126 expression in periodontal ligament cells increased expression of bone sialoprotein, osteopontin, and Runx2 among others through inhibitor of DNA binding 4 (ID4) [71]. Moreover, TMED and GPR126 are both associated with height in humans [72]. While *vegfab* was, like *gpr126*, enriched at 10 dpf in cluster 2, *id4* and *tmed10* were more broadly detected (Appendix A). The analysis of all differentially expressed genes with FishEnrichr [50,51] indicated that these genes are, for example, associated with the biological processes “sodium ion homeostasis”, “calcium ion homeostasis”, “response to osmotic stress”, and “response to ethanol” (Appendix A and Appendix A).

The analysis of *gpr126* expression in trunk sections of 3-months-old adult wildtype zebrafish revealed *gpr126* expression in both stromal and tubular mesonephric cells (Figure 6e), resembling the *Gpr126* expression pattern seen in mammalian embryonic kidneys (Figure 2).

## 4. Discussion

Our work indicates that *Gpr126* is involved in kidney development. *Gpr126* is detected in the pronephric and mesonephric kidneys of zebrafish, as well as in the metanephric kidneys of mice and humans. While *Gpr126* is widely expressed during development, its expression enriches in adulthood in parietal epithelial cells and epithelial cells of the collecting duct and the urothelium.

Most human aGPCR members and several orthologs have been detected in the kidney from zebrafish to human [8]. However, most of these data are based on RNA or “omics” approaches and only one member, Adgrc1, has been characterized in detail during renal development. Thus, it is crucial to explore strategies and methodologies aimed at providing a more detailed spatio-temporal analysis at cellular resolution. In this work, the RNAscope^®^ technology has been established to detect *Gpr126* at cellular resolution in the kidneys of zebrafish, mice, and humans during development and adulthood. The presented data show that RNAscope^®^ technology is suitable to detect *Gpr126* RNA in kidney FFPE sections and offers the possibility of combining in situ hybridization with immunohistochemistry on the same slide. Hence, RNAscope^®^ probes targeting *Gpr126* in combination with several kidney-specific antibodies served to conclude that *Gpr126* expression switches during metanephros development from ubiquitous to a tubule-enriched pattern in mice and humans, and remains low but consistently expressed in adult stages in the parietal epithelial cells and collecting ducts. In contrast, tubular epithelial collecting duct cells in the papillary region and the urothelial cells at the renal pelvis, ureter, and bladder show high *Gpr126* expression. Importantly, it has to be considered that the RNAscope^®^ technology detects only RNA levels and does not prove that the RNA will be translated into proteins or provides information such as subcellular localization of the protein, which would allow us to speculate on the function of the protein. Thus, it is important to continue to establish working anti-Gpr126 antibodies. 

In the mouse and human embryonic kidneys, the data presented here indicate that *Gpr126* is initially ubiquitously expressed in both mesenchymal and epithelial cell types, but ends up being enriched in the adult papillary collecting ducts and parietal epithelial cells. This suggests that *Gpr126* might be involved in differentiation of common early progenitor cells. There are other genes that show this behavior during kidney development, such as *PDGFRB* and *PECAM1*, glomerular markers of mesangial and endothelial cells, respectively. During human early kidney development, *PDGFRB* and *PECAM1* expressions are found in the renal mesenchyme but become enriched inside the glomeruli by 10 wg [73]. This might also explain why *Gpr126* is expressed in such different cell types, such as parietal epithelial cells and epithelial cells of the collecting duct and the urothelium, which to our knowledge exhibit no common function.

In the zebrafish, *gpr126* was detected in the pronephros and mesonephros. Similar to mammalian metanephros, *gpr126* expression was predominantly observed in tubular cells. However, in contrast to the mammalian kidney, some interstitial cells expressed very high levels of *gpr126*. Finally, *gpr126* was also found in the skin of zebrafish larvae, which is known to be a tissue rich in ionocytes. Ionocytes are found in the embryonic skin and adult gills of zebrafish. They express a variety of apical and basolateral cell surface transporters that control transepithelial movement of protons, sodium, potassium, ammonium, calcium, chloride, and bicarbonate [63,64]. Thus, ionocytes present analogous functions to collecting duct epithelial cells and their study might serve to investigate a possible role of *Gpr126* in ion homeostasis. In zebrafish, we found that *gpr126* is expressed in a sub-cluster of H^+^-ATPase-rich ionocytes. The epithelial transport functions of these cells are related to H^+^ secretion, Na^+^, and HCO_3_^−^ uptake, as well as ammonium excretion and are thus analogous to proximal tubular cells and type A-intercalated cells of the kidney in terms of transporter expression and function [65]. Ammonium secretion is mediated by *Rhcg1* (*rhcgb*) and *Rhbg*, which are both not detected in gpr126-expressing cells [65,74]. Apical acid secretion is largely mediated by H^+^-ATPase *atp6v1aa* and the sodium/hydrogen exchanger (*NHE3b/slc9a3.2*) [65,75]; expressions of both were detected in *gpr126*-expressing cells, whereby our data show that *slc9a3.2* expression is selectively detected in gpr126-expressing cells in cluster 2. Thus, *gpr126* is highly associated with apical acid secretion and might play a role in this process. Notably, gpr126 was also detected in type A-intercalated cells of the mouse kidney. Furthermore, the selective detection of the renal electrogenic Na^+^/HCO_3_^−^ cotransporter *NBCe1-A*/*slc4a4a* in *gpr126*-expressing cells indicates that *gpr126* might play also a role in HCO_3_^−^ uptake, contributing to the buffering of blood pH or intracellular pH [76]. Notably, *ECaC* (*trpv6*) expression was also selectively detected in *gpr126*-expressing cells. *ECaC* (*trpv6*) enables ionocytes to maintain high cytoplasmic Ca^2+^ concentration. *trpv6* encodes a constitutively open Ca^2+^ channel that functions as the first and rate-limiting step in the transcellular Ca^2+^ transport pathway [77]. In mouse kidney, Trpv6 has been described as being expressed at the apical domain of the distal convoluted tubules (DCT2), connecting tubules (CNT), and cortical and medullary collecting ducts (CD), whereby it co-localized with AQP2 in the CCD, OMCD, and IMCD [78] and thus shows a highly similar expression pattern to Gpr126 in mice. These data suggest that *gpr126* might also play a role in transcellular Ca^2+^ transport. This is supported by the co-detection of the cell-fate-related transcription factor *gcm2*, which regulates expression of the calcium sensor receptor gene *casr* [79]. The function of the novel marker of mature H^+^-ATPase-rich ionocytes, *ceacam1*, is currently unknown. Finally, *gpr126* was co-detected with the *Kir1.1* (*kcnj1a.1*) potassium transporter, also known as renal outer medullary K^+^ (*ROMK*). Its activity has been detected in the apical membrane of the thick ascending limb (TAL), DCT2, CNT, and CCD. Notably, *gpr126* was not detected in mice in TAL. However, while *kcnj1a.1* is important in the TAL for K^+^ recycling, which is essential for maintaining NaCl reabsorption via type II Na^+^-K^+^-2Cl-cotransporters, *kcnj1a.1* plays an important role in mediating K^+^ excretion in the DCT2, CNT, and CCD [80,81]. Collectively, the zebrafish ionocyte data fit with the observed *Gpr126* kidney data in mice.

The analysis of co-detection of genes together with *gpr126* suggests that Gpr126 plays a role in the different functions described above. On the other hand, Gpr126 might be required for the maturation of ionocyte progenitor cells into a distinct sub-population of ionocytes. Future experiments modulating *Gpr126* expression followed by functional analysis will reveal the role of Gpr126 in heart development and physiology.

Notably, the collecting ducts consist of principal and intercalated cells (similar to H^+^-ATPase-rich ionocytes), but the colocalization analyses revealed that *Gpr126* is not restricted to either cell type. *Gpr126* is highly detected throughout the collecting duct epithelial cells of the medullary papilla and in the urothelial cells of the renal pelvis, ureter, and bladder. Notably, inner medullary and papillary collecting ducts are continuously subject to osmotic and hydrostatic stress. They collect the hyperosmotic filtrate from multiple nephrons and withstand a fluid shear stress that oscillates between 0.2 and 20 dyn/cm^2^, which is equivalent to the physiological range of blood shear stress in resistance vessels [82]. The urothelial cells of the renal pelvis, ureter, bladder, and proximal urethra receive the final urine and are also reported to respond actively to mechanochemical changes [83] in many cases through GPCRs [84,85]. *Gpr126* has been detected in a variety of tissues and cell types, including vascular endothelial cells, subject to mechanical stress [17] and has also been described as a mechanosensor that responds to signals in the extracellular matrix by binding with type VI collagen [36] and laminin-211 [35]. This supports the idea that *Gpr126* could act as a mechanosensitive gene in the epithelium of the urinary collecting system: collecting ducts, renal pelvis, ureters, bladder, and urethra, and possibly also parietal epithelial cells.

## 5. Conclusions

The results shown in this work demonstrate that *Gpr126* is expressed in the kidney during development and adulthood, while its expression pattern is evolutionarily conserved from zebrafish to human. We provide a detailed expression analysis of *Gpr126* and identified several associations to possible functions of *Gpr126* in the kidney as a basis to elucidate a possible role of Gpr126 in kidney development and/or physiology. Notably, database analyses associated increased Gpr126 expression also with kidney disease [8]. Thus, in the future, it will be important to characterize how *Gpr126* expression is altered in kidney disease and to determine if *Gpr126* deletion affects kidney development and/or physiology.

## Figures and Tables

**Figure 1 cells-12-01988-f001:**
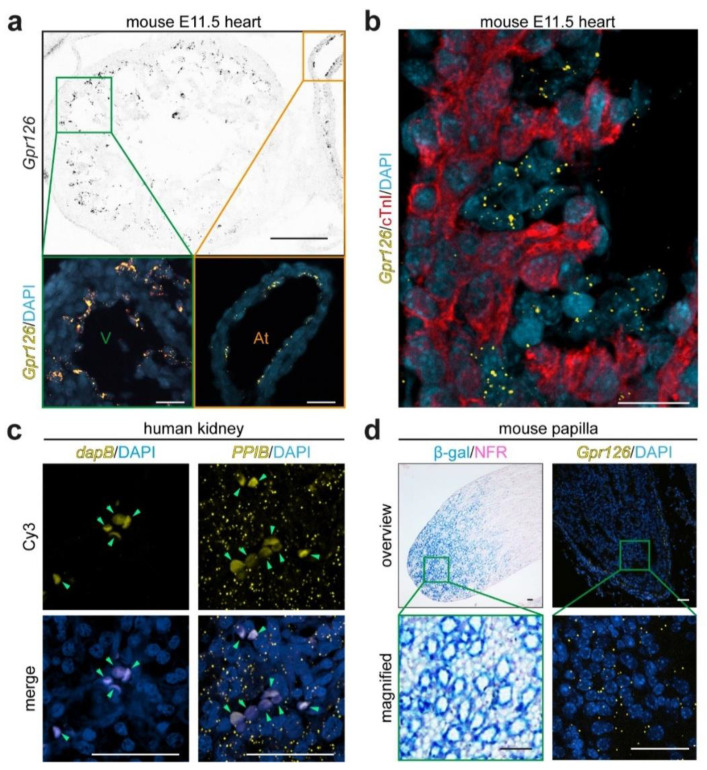
Validation of RNAscope^®^ for mouse and human kidneys. (**a**) *Gpr126* mRNA-targeting RNAscope^®^ probes were used on transversal sections of FFPE mouse wildtype embryonic hearts at 11.5. Signal (yellow dots) was found lining the ventricular (V) and atrial (At) chambers. DAPI was utilized to visualize nuclei. Black scale bar: 100 µm; white scale bars: 25 µm. (**b**) RNAscope^®^ technology was combined with anti-cardiac troponin I (cTnI) antibody staining to identify the myocardium (red). *Gpr126* signal was localized in the cTnI-negative endocardial cells of mouse embryonic hearts at E11.5. Scale bar: 25 µm. (**c**) RNAscope^®^ probes targeting the bacterial *dapB* mRNA and the human *PPIB* mRNA were used on human FFPE renal tissues as negative and positive technical controls, respectively. Signal detection (yellow dots) was performed using a dye with excitation and emission light properties equivalent to those of cyanine 3 (Cy3). Note: when present, red blood cells exhibit autofluorescence and are seen as bigger corpuscles (green arrowheads). Scale bars: 50 μm. (**d**) *Gpr126-lacZ* reporter mouse line revealed β-galactosidase activity (β-gal, blue) in the adult renal medullary papilla upon X-gal staining on a nuclear fast red (NFR, pink) background staining. This papillary expression pattern was confirmed by fluorescent RNAscope^®^ in situ hybridization (yellow dotty signal) in adult FFPE kidney sections. Scale bars: 50 μm.

**Figure 2 cells-12-01988-f002:**
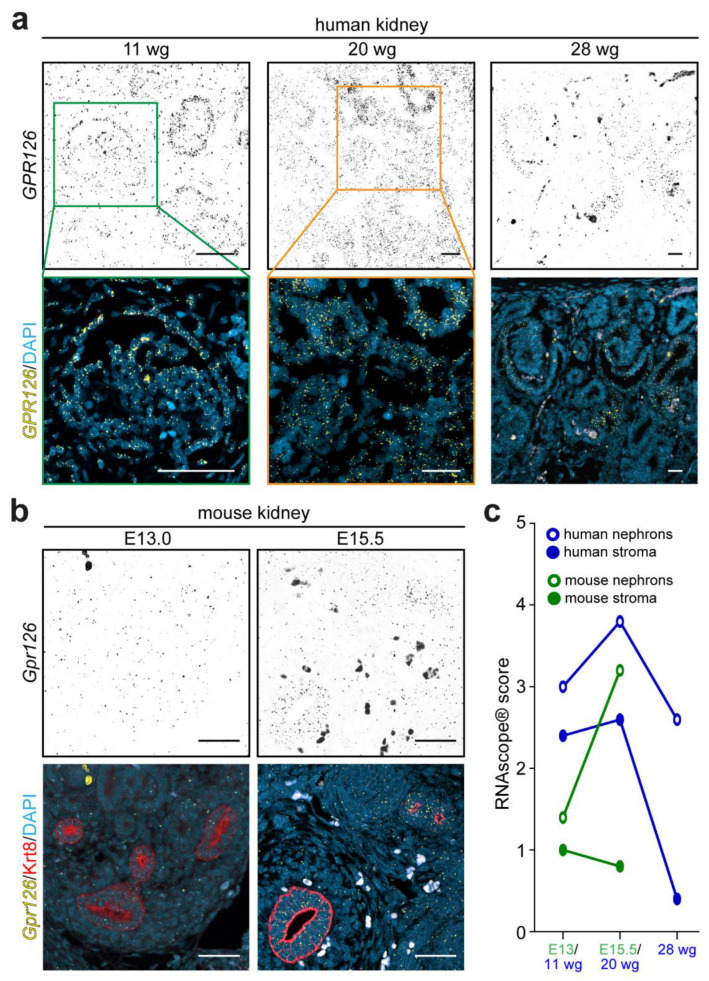
*Gpr126* expression during metanephric development. (**a**) RNAscope^®^ probes targeting the human *GPR126 m*RNA were used on human FFPE embryonic metanephric kidney sections at 11, 20, and 28 weeks of gestation (wg). Over time, the *GPR126* expression pattern (yellow dotty signal) concentrates in the developing nephrons, compared to stromal cells. DAPI was utilized in (**a**,**b**) to visualize nuclei. (**b**) RNAscope^®^ probes targeting the mouse *Gpr126* detected low, but ubiquitously distributed, signal (yellow dotty signal) in the mouse FFPE-developing metanephros sections at E13.0 compared to samples obtained at E15.5, where signal was found more localized in cytokeratin-8-positive ureteric bud cells (red). Scale bars: 100 μm. (**c**) Representation of the RNAscope^®^ signal score means obtained from epithelial (circumferences) and stromal (circles) cell types of human (blue) and mouse (green) developing metanephroi during the physiological nephrogenesis burst window (20 wg in human, n = 5; E15.5 in mouse, n = 3), earlier (11 wg in human, n = 4; E13.0 in mouse, n = 3), and later (28 wg in human, n = 3) stages.

**Figure 3 cells-12-01988-f003:**
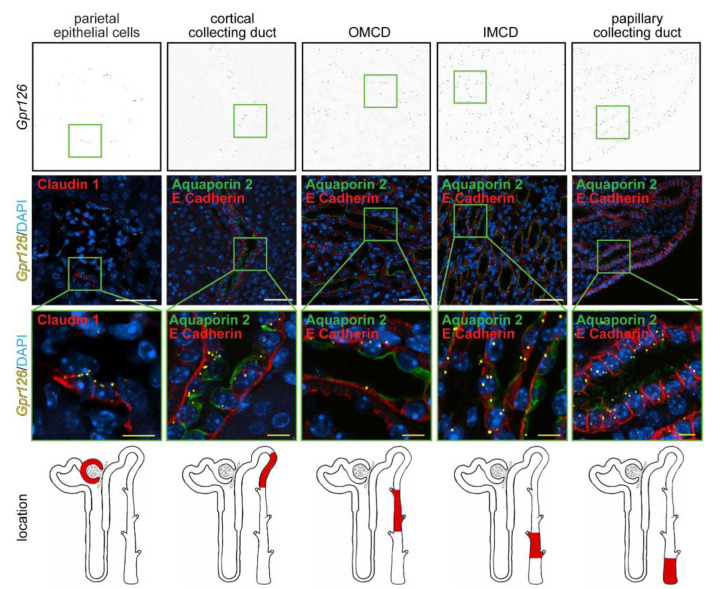
Renal expression of *Gpr126* in adult mice. RNAscope^®^ probes targeting the murine *Gpr126* mRNA (yellow dots) were combined with renal-specific immunostaining (red) on FFPE kidney tissue of wildtype C57BL/6 mice. *Gpr126* signal colocalized the claudin 1-positive parietal epithelial cells lining the Bowman´s capsule and the aquaporin 2/E-cadherin-positive collecting duct. Note *Gpr126* expression in the cortical, inner (IMCD), and papillary segments of the collecting duct is enriched compared to the outer segment (OMCD). DAPI was utilized to visualize nuclei. White scale bars: 50 µm. Yellow scale bars: 10 µm.

**Figure 4 cells-12-01988-f004:**
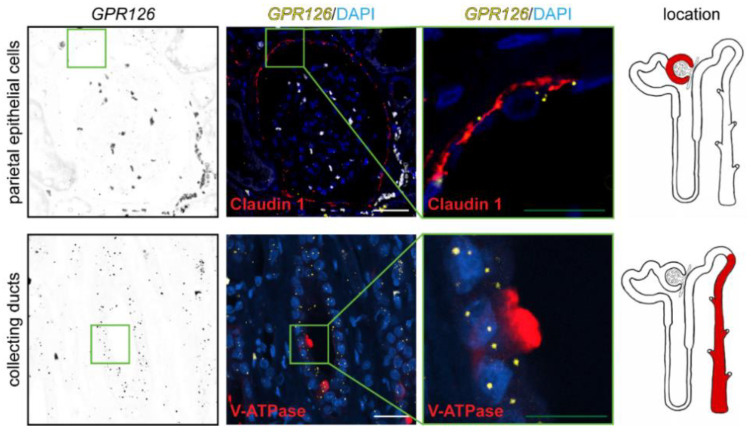
*GPR126* expression in the adult human kidney. RNAscope^®^ probes targeting the human *GPR126 m*RNA were used on human FFPE renal tissues in combination with antibodies serving as renal-specific markers. *GPR126* signal (yellow dots) colocalized the parietal epithelial cell marker claudin 1 and the collecting duct epithelial cell marker V-ATPase (markers in red). DAPI was utilized to visualize nuclei. White scale bars: 50 μm. Green scale bars: 20 μm. The red color in the nephron schemes depicts the specific locations where the respective markers are expected to be found.

**Figure 5 cells-12-01988-f005:**
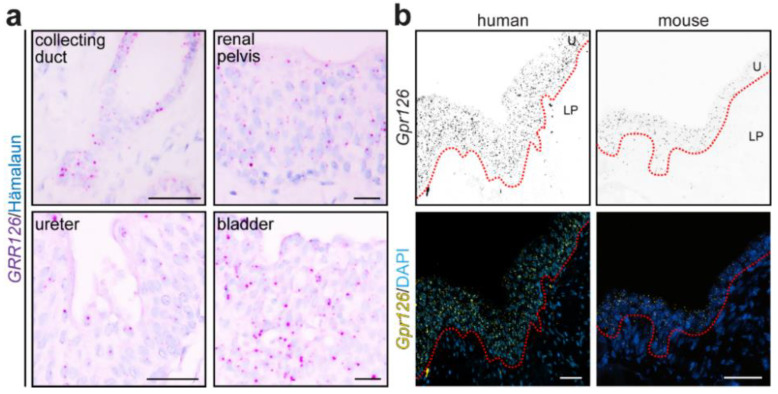
*Gpr126* is expressed throughout the urinary collecting system. (**a**) RNAscope^®^ probes targeting the human *GPR126 m*RNA were used on human FFPE urinary tissues. *GPR126* signal (magenta dots) was found in the epithelial compartment of the urinary collecting system, collecting duct epithelial cells and transitional epithelium (urothelium) of the renal pelvis, ureter, and bladder. Hematoxylin was utilized to visualize nuclei. Scale bars: 20 μm. (**b**) RNAscope^®^ probes targeting *Gpr126 m*RNAs were used on FFPE renal pelvis sections of human and mouse. Both species showed *Gpr126* enrichment in the urothelium (U) of the renal pelvis mucosa (yellow dots), contrary to the lamina propria (LP) layer beneath. DAPI was utilized to visualize nuclei. Dashed red line represents the approximate location of the basement membrane at the urothelium–lamina propria junction. Scale bars: 50 μm.

**Figure 6 cells-12-01988-f006:**
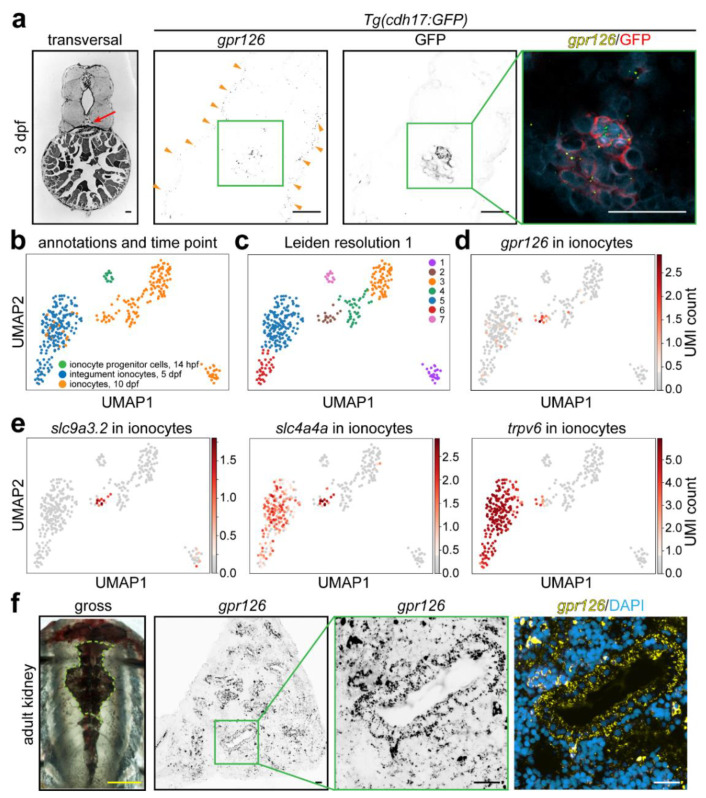
*gpr126* expression in zebrafish kidneys and ionocytes. (**a**) RNAscope^®^ probes targeting the zebrafish *gpr126* RNA were used on FFPE transversal sections of a renal tubule-specific reporter zebrafish line at 3 days post fertilization (dpf). Red arrow: pronephros. Orange arrowheads: ionocytes. Bright-field picture taken from ZFIN Atlas of Zebrafish Anatomy. (**b**–**e**) UMAP plot of single-cell RNAseq data. (**b**) Ionocytes are represented with false coloring, showing ionocytes progenitor cells at 14 h post fertilization (hpf, light green), integument ionocytes at 5 dpf (blue), and ionocytes at 10 dpf (orange). Cell types were annotated in the original study. (**c**) Leiden clustering with a resolution of 1. (**d**) UMAP plot of *gpr126* in ionocytes, whereby the normalized and scaled UMI (Unique Molecular Identifier) counts are colored in a scale from 0 UMI counts (gray) to the highest UMI count (dark red). (**e**) UMAP plot as described in (**d**), depicting s*lc9a3.2*, *slc4a4a*, and *trpv6* in ionocytes. (**f**) *gpr126* signal in the adult (3 months) zebrafish kidney indicates high expression in a subset of tubular and non-tubular cells. Green dashed line: kidney trunk region, from which sections were obtained. Scale bars: yellow: 0.5 cm, black and white: 25 µm.

## Data Availability

The data are all included in the article and Appendix A. No omics data were generated.

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
