# Peer review of "Adhesion GPCR Gpr126 (Adgrg6) Expression Profiling in Zebrafish, Mouse, and Human Kidney"

_cells, 2023, doi:10.3390/cells12151988_

Round 1

Reviewer 1 Report

The authors of the submitted manuscript with the following title “Adhesion GPCR Gpr126 (Adgrg6) Expression Profiling in Zebrafish, Mouse and Human Kidney” conducted an interesting study investigating the expression patterns of GPR126.

The authors found, that GPR126 mRNA can be detected in pronephric and mesonephric kidneys in zebrafish and metanephric kidneys in human and mice. An increase in mRNA levels of GPR126 can be found in adulthood in parietal epithelial cells and epithelial cells of the collecting duct and urothelium.

The findings allow the hypothesis that GPR126 could be involved in kidney development. For further validation of this hypothesis, detailed studies are needed. Nevertheless, the characterization of mRNA levels of GPR126 in the different developmental stages of the kidney generate the foundation for future studies.

Overall, the manuscript is well written. The introduction as well as materials and methods provide all information necessary. The results are good to follow, and the figures are of high quality. The discussion is critical and addresses the weaknesses of the study as well as the necessity.

Minor changes:

-          Line 21:“aGPCRs play an important role in development and disease and have recently been associated to the kidney”, phrasing is vague and therefore misleading. How associated to the kidney? High expression or pathophysiological?

-          Line 22:“How aGPCRs function and what their roles are in the kidney are poorly understood”, rephrasing (suggestion: general aGPCR function and their physiological role in the kidney is poorly understood)

-          Line 51: citation missing, PMID: 25533341

-          Line 89-92: Reference 30 from 2004. Please include GRKs and arrestins.

-      Line 141 Hybridization, hybridization

Author Response

We thank the reviewer for the very positive review. The suggested minor changes have been addressed as follows:

- Line 21:“aGPCRs play an important role in development and disease and have recently been associated to the kidney”, phrasing is vague and therefore misleading. How associated to the kidney? High expression or pathophysiological? To address this issue, we have added the following sentence, which is a short summary of the second part of the introduction: “Several aGPCRs are expressed in the kidney and some aGPCRs are either required for kidney development or their expression level is altered in diseased kidneys.”

- Line 22:“How aGPCRs function and what their roles are in the kidney are poorly understood”, rephrasing (suggestion: general aGPCR function and their physiological role in the kidney is poorly understood) We have rephrased the sentence as follows: “Yet, general aGPCR function and their physiological role in the kidney are poorly understood.”

- Line 51: citation missing, PMID: 25533341 Thank you for indicating this point. As recommended, we have added the reference.

- Line 89-92: Reference 30 from 2004. Please include GRKs and arrestins. After re-reading the indicated reference as well as checking the literature for “Gpr126 or Adgrg6” combined with “arrestin or GRK”, we could not find any indication that there is a relationship. The only publication found (PMID: 35394864) states: “The results showed no obvious Gq activity or b-arrestin-2 recruitment of GPR126 in response to progesterone or 17OHP stimulation, using angiotensin II and AT1R as controls (SI Appendix, Fig. S3 G and H) (45, 46). Collectively, these results indicated that progesterone selectively induces the Gi coupling of GPR126 but is not detectable with Gs, Gq, or b-arrestin-2, whereas 17OHP induces both Gs and Gi coupling of GPR126 but not Gq or b-arrestin-2 coupling”. Therefore, we have not altered the text. If the reviewer provides distinct literature, it will be our pleasure to include this information.

- Line 141 Hybridization, hybridization The phrase “For RNAscope® In Situ Hybridization,” has been changed to “For RNAscope® in situ hybridization”.

Reviewer 2 Report

This is a very well done descriptive manuscript on the localization of mRNA for Gpr126 in human, mouse and zebra fish kidney.  A combination of RNA scope and immunofluorescence techniques enabled specific characterization of expression. profiles of Gpr126. Taken together the authors show that mRNA  expression. profile for  Gpr126 is conserved in these species. Localization in pronephric and mesonephric kidneys  in zebra fish as well as mesonephric kidney in mouse na humans during development suggesting a role for this protein in development. In adults Gpr126  in parietal epithelial cells and epithelial cells of  the collecting duct and the urothelium.

Additionally in Zebra fish GRP126 was found in skin ionocytes and suggest a role in controlo of ion transport.

Annalysis of single cell expression. data confirms this results. Further analysis of the single cell data could indicate whether other aGPCRs  are expressed in the same cluster of cells or if this is specific for GPR126. Furthermore, it would be interesting to see if regulatory elements for GRP126 and other genes express in the same clusters are conserved.

As discussed by the authors RNA expression does not prove the protein is expressed and further studies will have to be performed to confirm the localizaation/ cell types expressing GPR126.

Author Response

We thank the reviewer for the very positive review. The reviewer did not clearly indicate if a revision is requested but we decided to address the following comment:

Analysis of single cell expression. data confirms this results. Further analysis of the single cell data could indicate whether other aGPCRs are expressed in the same cluster of cells or if this is specific for GPR126. Furthermore, it would be interesting to see if regulatory elements for GRP126 and other genes express in the same clusters are conserved.

To address this comment, we have provided additional data:

1) Table S2 (Ionocyte expression of aGPCRs) and Figure S2a,b (gpr126 co-expression analysis) which show that the only aGPCR expressed in a similar manner as gpr126 is adgrl3. In addition, we added the following text: “To determine whether other aGPCRs exhibit a similar expression pattern as gpr126 in ionocytes, we analyzed all genes starting with “adgr” (Table S2). This analysis revealed that only a few aGPCRs are enriched in a specific cluster (Table S2 and Figure S2a). adgrl3 was the only aGPCR found that exhibits a similar pattern as gpr126 (Figure S2a).”

2) Figure S2c and Table S3 (Differentially expressed genes comparing cells in which gpr126 was detected versus cells that lack gpr126 UMIs) and Table S4 (FischEnrichr analysis of differentially expressed genes). In addition, we added following text: “In addition, we performed a differential expression analysis, where we compared cells in which gpr126 was detected versus cells that lack gpr126 UMIs (Table S3). Among the differentially expressed genes, we detected 3 genes that have previously been associated to Gpr126 (Figure S2b). It has been described that GPR126 regulates angiogenesis through modulation of VEGFR2 receptor signaling [70]. Further, GPR126 expression in periodontal ligament cells increased expression of bone sialoprotein, osteopontin, and Runx2 among others through inhibitor of DNA binding 4 (ID4) [71]. Moreover, TMED and GPR126 are both associated with height in humans [72]. While vegfab was like gpr126 enriched at 10 dpf in cluster 2, id4 and tmed10 were more broadly detected (Figure S2a). The analysis of all differentially expressed genes with FishEnrichr [50,51] indicated that these genes are for example associated with the Biological Processes “sodium ion homeostasis”, “calcium ion homeostasis”, “response to osmotic stress”, and “response to ethanol” (Figure S2c and Table S4)”.

3) The Method section and the listing of “Supplementary Materials” was modified accordingly.